# Design and Evaluation of a Pediatric Lower-Limb Exoskeleton Joint Actuator

**Anthony Goo [1], Curt A. Laubscher [1], Ryan J. Farris [2] and Jerzy T. Sawicki [1,*]**

[1]   Center for Rotating Machinery Dynamics and Control (RoMaDyC), Washkewicz College of Engineering, Cleveland State University, Cleveland, OH 44115, USA; accg1994@gmail.com (A.G.); laubscher.curt@gmail.com (C.A.L.)

[2]   Human Motion and Control Division, Parker Hannifin Corporation, Macedonia, OH 44056, USA; ryan.farris@parker.com

[*]   Correspondence: j.sawicki@csuohio.edu

**Abstract:** Lower-limb exoskeletons have undergone significant developments for aiding in the ambulation of adults with gait impairment. However, advancements in exoskeletons for the pediatric population have comparatively been lacking. This paper presents a newly developed joint actuator designed to drive the hip and knee joints of a pediatric lower-limb exoskeleton. The performance requirements associated with the actuators were determined based on a target audience of children ages 6–11 years old. The developed actuators incorporate a hybrid belt-chain transmission driven by a frameless brushless DC motor. One actuator underwent benchtop testing to evaluate its performance with respect to their torque production, bandwidth properties, backdrivability in terms of inertia and friction characteristics, speed capabilities, and operational noise levels. As a preliminary validation, a set of actuators were placed in a prototype orthosis to move a pediatric test dummy in gait tracking via state-feedback control. The results showed that the newly developed actuators meet the design specifications and are suitable for use in the pediatric exoskeleton being developed.

**Keywords:** lower limb exoskeletons; actuator; design; pediatric; backdrivability; validation

## 1. Introduction

Gait impairment can arise in children suffering from developmental, neuromuscular, or neurological disorders such as cerebral palsy (CP), muscular dystrophy (MD), and spina bifida (SB). CP refers to a group of disorders that affect movement and posture due to disturbances that occurred in the developing fetal or infant brain [1]. According to the Centers for Disease Control, CP is the most common source of gait impairment in the pediatric population, at a prevalence rate of about 3.1–3.6 per every 1000 children [2–5]. An estimated 8000 new cases are reported each year [5], causing a wide variety of gait impairments [6,7]. MD is a genetic disorder which commonly causes gait impairments due to the progressive loss of muscle fibers [8]. Some common forms of MD, such as Duchenne or Becker muscular dystrophy, affect approximately 2 in every 10,000 boys aged 5 to 9 years old in the United States [9]. Patients are expected to lose ambulation capabilities by the time they are 9 years old [10]. The last of the representative disorders, SB, affects 3.1 out of every 10,000 children up to age 19 in the United States [11]. SB, also referred to as myelodysplasia, is a developmental disorder in which the spinal cord does not fully seal. The disorder can manifest as myelomeningocele, which is the development of a sac of fluid protruding from the back containing a part of the damaged spinal column and nerves. This results in decreased ambulatory functions, the severity of which is dependent on the location of the spinal cord defect [5].

Taken together, the pediatric patients afflicted by these disorders require intervention to improve, prolong, or restore ambulation and mobility. Customary approaches for gait rehabilitation include physical therapy and bodyweight supported treadmill training (BWSTT) [12]. Powered lower-limb orthoses, or lower-limb exoskeletons, have been developed to address the rehabilitation and assistance of those suffering from gait impairments [12–14]. An exoskeleton is a wearable robotic device capable of providing joint torque to control or assist user motions. Exoskeletons address the limitations present in these traditional methods by relieving physical therapists of the physical work associated with manual methods [13] and increasing the repeatability of the walking task via robotic accuracy [15]. As a partial byproduct of the reduced physical burden on therapists, patients often have more productive and efficient therapy sessions. Furthermore, exoskeletons can be designed to allow mobility and provide walking assistance when used outside therapy sessions.

Although there has been significant development in adult exoskeletons, the pool of options that serve the pediatric population is comparatively smaller [16]. Of those that exist, most are not wholly suitable for assisting a patient suffering from gait impairment in functional mobility. The Lokomat (Hocoma Inc.) is an immobile BWSTT system with an attached robotic exoskeleton which augments patient movements at both the hip and knee joints [15,17,18]. Originally developed for adult patients, the system has been adapted for the gait training of children with CP [18,19]. The Lokomat, and other stationary systems such as the WALKBOT [20,21] and one developed by Giergiel et al. [22], are limited in application due to their stationary nature. Similarly, devices affixed to large mobility aids, such as the CPWalker [23], present limitations to use in confined spaces and in community settings. The ATLAS 2020 (Marsi Bionics S.L.) is a pediatric device with 10 actuated degrees of freedom (DOF) for children ages 3 to 12 years old [24–26]. Due to the large number of actuators and DOF, the device is bulky and heavy, weighing approximately 12 kg [24]. Conversely, some powered lower-limb orthoses are designed to support only a single DOF. Example exoskeletons actuating only the knee include a device created by Copilusi et al. [27,28] and the devices created and tested by the National Institute of Health (NIH) [29,30]. Two additional exoskeletons from the literature appear to no longer be under active development, including the WAKE-up exoskeleton [31,32] and a device presented by Canela et al. [33]. A publication has presented a new pediatric exoskeleton, P-LEGS [34]. The 8 kg orthosis actuates the hip, knee, and ankle joints in the sagittal plane for users of 1.00–1.23 m in height and 16–28 kg in weight. The device shows promise but has yet to be clinically evaluated. Thus, there remains a need for lower-limb exoskeletons designed for the pediatric population without the aforementioned limitations.

Our group at Cleveland State University (CSU) has begun the development of the CSU Pediatric Exoskeleton to improve functional mobility in children suffering from gait impairments. The creation of such a device requires the design and testing of an actuator suited for this purpose. Previously, our group developed an original actuator design which underwent benchtop evaluation [35,36]. The actuators have also been experimentally tested for crouch-to-stand motion [37], gait guidance [38], and assistive capabilities [39]. In these experiments, various issues were encountered, and hardware limitations were revealed. What follows in this paper is the redesign of the actuator based on experience gained from the design, fabrication, and evaluation of the previous version of the actuator. Section 2 discusses the actuator performance requirements based on pediatric gait and reviews the actuator designs used in other pediatric exoskeletons. This information is used to generate a list of design specifications for our new prototype actuator. The finalized design and the basic characteristics of this actuators are discussed in this section. Section 3 details experimental benchtop evaluations for the designed actuator, entailing torque and bandwidth tests, backdrivability quantification in terms of friction and inertia, and operational noise and speed tests. Section 4 includes a preliminary evaluation of the actuators via state-feedback gait tracking on a prototype orthosis and pediatric test dummy. Section 5 discusses the conclusions drawn from the experimental evaluations, as well as remarks on prospective design considerations and future work.

## 2. CSU Pediatric Exoskeleton Actuator

### 2.1. Performance Requirements for Pediatrics

For use in an assistive pediatric exoskeleton, the actuator must meet a certain set of performance requirements. These requirements are primarily derived from information about nominal pediatric gait and the target population. The pediatric exoskeleton is designed with a target age range of 6–11 years old. The lower bound of this age range was determined as a practical matter; the patient would have needed to develop essential motor skills as well as cooperating with the researchers. Children will begin to display proficiency in functional mobility skills by the time they are 5–7 years old [40,41]. With respect to cooperation with researchers, studies suggest that by age 5, children are able to meaningfully concentrate on a given task for about 14 minutes at a time [42], increasing at a rate of about 3–5 minutes per year in young children [43]. The designation of the upper limit was determined based on anthropometric considerations. Census data suggests that children 11 years old will have a height of 150 cm on average [44], placing them on the cusp of graduating to preexisting devices that exist for adults. For reference, the Indego Therapy device is suited for users 155–191 cm in height [45]; the EksoNR device by Exo Bionics has suggested height requirement between 153–193 cm [46]; and the Rex Bionics device is designed for patients between 142–193 cm in height [47].

The pediatric exoskeleton is designed such that it provides torque at both the hip and knee joints. The choice to exclude ankle actuation was made observing that the Indego Therapy exoskeleton [48] and ReWalk exoskeleton [49] have been clinically demonstrated to provide useful gait assistance to individuals with a spinal cord injury without active ankle actuation. Thus, to reduce the overall weight and complexity of the device, we have elected to keep the ankle free. In practice, if a therapist deems ankle support is necessary for the patient, an ankle foot orthosis may be used in conjunction with the exoskeleton. In addition, the hip and knee joint actuators are intentionally identical and modular to simplify design considerations and improve serviceability.

With the age range established, information about the expected joint trajectories and torque profiles can be extrapolated, which in turn defines the performance requirements for the actuators. CP is a heterogenous disorder which has a large variability in how it affects a patient's gait [2]. Thus, the device was designed to help patients of various levels of impairment to achieve a nominal healthy gait. The joint angle measurements of pediatric gait tend towards adult-like values by age 4 [50]. Thus, although children tend to display higher gait cadence, an adult-like cadence of about 107 steps per minute will be assumed [51]. Based on anthropometric data from census surveys [44], the assumed cadence, and normalized gait data provided by Winter [51], the expected gait statistics were calculated for an 11-year-old to determine the upper limit of performance required by the actuator. The relevant gait information for the hip and knee joints determine the actuator performance requirements, which are shown in Table 1, and include range of motion, joint velocity, peak torque, root-mean-square (RMS) torque, and maximum power. For clarification, flexion of a joint is associated with positive values. The actuator performance requirements should be set such that it is able to sweep at least the full range of motion, have a nominal output speed exceeding the expected peak joint velocity, achieve a maximum output torque at least that of the peak torque, reach a continuous torque comparable to the RMS torque, and have a power rating greater than that observed in gait. An actuator greatly exceeding these would be larger and heavier, so compromise must be made to keep size and weight low while still providing adequate assistance, which is especially important for the pediatric population.

**Table 1.** Performance requirements for a 1.1 s gait period.

| Performance Requirement | Hip | Knee |
|---|---|---|
| Range of Motion (deg) | −11 to 22 | 0 to 65 |
| Joint Velocity (deg/s) | −80.7 to 156.6 | −361.6 to 305.8 |
| Peak Torque (Nm) | −19.6 to 29.1 | −29.8 to 13.6 |
| RMS Torque (Nm) | 11.9 | 14.3 |
| Maximum Power (W) | 34.6 | 47.3 |

## 2.2. Prior Work on Pediatric Exoskeleton Actuators

The pediatric exoskeletons described in Section 1 utilize various motors and transmission mechanisms in their joint actuators to provide joint torque. In this section, reported information on these actuators is briefly summarized in Table 2.

**Table 2.** Summary of reported information on existing actuators for pediatric exoskeletons.

| Exoskeleton | Motor | Transmission Type | Cont./Peak Toque | Velocity | Source |
|---|---|---|---|---|---|
| Lokomat | Maxon RE 40 | Belt drive and ball screw | 30/160 Nm at knee50/280 Nm at hip | – | [15,17–19] |
| Walkbot-K | – | – | –/– | – | [20,21] |
| Giergiel et al. | Servo motor (HiTEC HS 805 BB) | Cyclo gear reduction | –/– | – | [22] |
| CPWalker | 24 V 100 W brushless DC motor (Maxon EC-60) | Harmonic drive (160:1) | 36.3/669 Nm * | 144 deg/s * | [23] |
| ATLAS 2020 (ARES actuator) | 90 W brushless DC motor (Maxon) | Harmonic drive (100:1) | –/76 Nm | – | [24–26] |
| Copilusi et al. | – | Cable driven knee | –/– | – | [27,28] |
| Lerner NIH | 24 V 90 W brushless DC motor | Planetary gear (89:1) Chain sprocket (3.5:1) | 16.1 Nm/– | – | [29,30] |
| Wake-up | Servo motor (Dynamixel EX-106+) | Pulley based reduction (1.5:1) | –/6 Nm | 400 deg/s | [31,32] |
| Canela et al. | 70 W brushless DC motor (Maxon EC-45) | Harmonic drive (160:1) | 20.5/233 Nm * | 229 deg/s * | [33] |
| P-LEGS | 24 V Maxon motor | Geared reduction (160:1) | 13.5/76 Nm | – | [34] |
| Laubscher et al. | 24 V brushless DC motor | Belt transmission (40.6:1) | 5.4/35.7 Nm | 375 deg/s | [35,37–39] |

– Information not reported. * Derived from rated motor parameters and the cited transmission ratio.

Although many devices have kept their actuator information undisclosed, some observations can be made. Firstly, electrically powered actuators appear to be a common choice for the mode of power transfer in exoskeleton actuator design. This is advantageous over other modes of power transfer such as hydraulics or pneumatics for a number of reasons. Electrical power is reliable and convenient for implementation because motors are readily available on the market and are relatively small and lightweight. Power can be easily sourced externally through the mains or a battery included on the exoskeleton. Hydraulic and pneumatic systems are typically tethered to an external compressor or pump which can be cumbersome and noisy, making them impractical to use outside a clinic. Additionally, pressurized fluids pose safety risks for the user and bystanders, an important consideration for an exoskeleton for pediatrics. Secondly, some of the presented actuators are not modular because they are built into the exoskeleton. This can be undesirable, because modularity improves ease of maintenance, making it possible to quickly swap out dysfunctional actuators when servicing the exoskeleton. In a clinical setting, this maximizes functional exoskeleton uptime for children in therapy. Lastly, many of the transmission types inherently have high impedance, reducing the possibility of backdrivability in the joints of the exoskeleton. Backdrivability is a desirable characteristic in exoskeleton actuators because it promotes patient participation and comfort [52] and is essential in rehabilitation robotics [53]. The actuator ought to provide adequate torque to achieve meaningful assistance while still being backdrivable to maximize usability and safety. Some of the actuators have series elastic qualities due to the incorporation of various springs in the joint design. This is to enable

compliant behavior in the actuated joint in an otherwise non-backdrivable system. Examples include the ARES actuator joint on the ATLAS 2020 [26] and the WAKE-up exoskeleton [31].

The actuator described in this paper was specifically developed to be electrically powered, modular, and backdrivable for the reasons mentioned above. The design is a revision of a previous version of the actuator [35,36]. The previous 0.6 kg actuator utilized a 70 W brushless DC motor with a 40.6:1 belt-based transmission to achieve 5.4 Nm and 35.7 Nm of continuous torque and peak torque, respectively. The novel actuator presented in this paper aims to improve backdrivability characteristics and to provide increased power density by improving torque capabilities while decreasing the size and weight.

### 2.3. Design Specifications and Hardware Description

Based on the target users discussed in Section 2.1 and the remarks on other actuators in Section 2.2, a list of design specifications was formulated and is summarized in Table 3. An actuator satisfying these design specifications should be able to assist the majority of the target population in achieving a nominal healthy gait.

**Table 3.** Summary of the design specifications for the pediatric exoskeleton joint actuator.

| Category | Specification |
|---|---|
| Range of motion | Sweeps −11 to 22 deg at the hip and 0 to 65 deg at the knee without hyperextension |
| Velocity | Achieves joint velocities of at least 160 deg/s at the hip and 370 deg/s at the knee |
| Torque | Supplements joint torques up to, or in excess of, 30 Nm at both the hip and knee |
| Power | Provides power to both joints of up to, or in excess of, 47.3 W driven by electric motors |
| Modular design | Actuator can be inserted into exoskeleton for driving either hip or knee joint |
| Backdrivable | Low impedance achieved through minimal friction and inertia characteristics |
| Sound | Operates quietly with low noise levels |
| Weight | Weighs no more than 0.6 kg |
| Size | Dimensions comparable to or smaller than $46 \times 79 \times 160$ mm |

The new actuator design is driven by a 144 W brushless DC motor (Megaflux MF0060008-48V), supplying more power than the previous actuator. Torque is delivered through a hybrid belt–chain transmission with a speed-reduction ratio of 20.4:1, as shown in Figure 1. This produces quiet and robust actuator performance with a continuous output torque of 5.9 Nm, a theoretical peak output torque of 46.9 Nm, and an unloaded output speed of 1300 deg/s. The 0.45 kg actuator, depicted in Figure 2, is fully appropriate for use in pediatric exoskeletons given the design specifications. The motors are controlled using servo-amplifiers (Maxon ESCON 50/5) and dSPACE MicroLabBox, which use Hall effect sensors and magnetic angle sensors to provide velocity and position feedback.

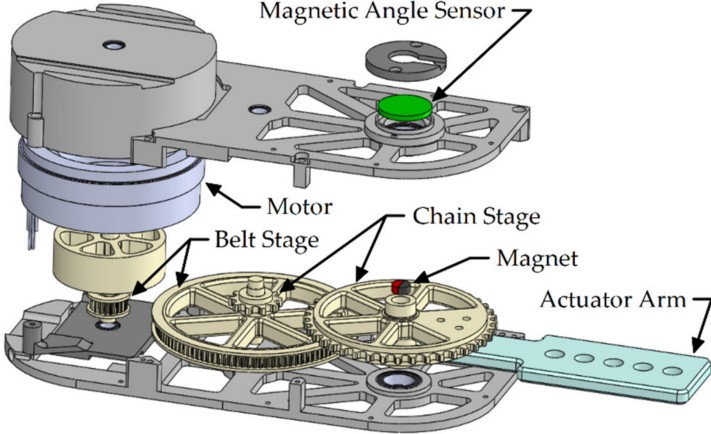

**Figure 1.** Exploded view of the joint actuator design for the pediatric exoskeleton.

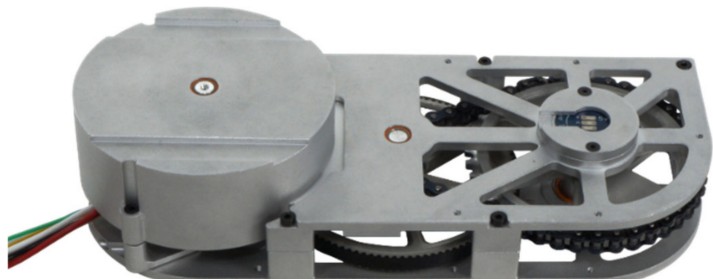

**Figure 2.** Picture of joint actuator for the pediatric exoskeleton configured for free-spinning without an actuator arm.

## 3. Actuator Benchtop Testing

To test the capabilities of the design, one actuator was experimentally evaluated in various benchtop settings. Firstly, the torque capabilities were established and the bandwidth was determined. Secondly, backdrivability was assessed in terms of inertia and friction in the system. Thirdly, the actuator was tested for its speed capability and associated operating noise. For a point of comparison, a number of the experiments were compared to the original version of the actuator to show points of improvement or regression.

### 3.1. Torque and Bandwidth Response

A static torque response test was performed to evaluate how well the actuators can follow commanded torques when there is little motion. This was accomplished by applying a known load on the actuator arm with the actuator providing an opposing torque, with the experimental setup shown in Figure 3. For each applied load, a commanded current supplied from the servo-amplifier was gradually increased until the actuator arm remained suspended in place. The results are presented in Figure 4, and include two experiments. The primary test supplied current for approximately 1 s and used small increments in the applied load to achieve fine gradations. The experiment was halted at higher torque levels due to an observed increase in motor temperature. A high load test was separately conducted for evaluating higher load limits of the actuator with current duration less than 0.5 s. The results showed that an experimentally tested peak torque of 21.1 Nm was possible. Additionally, there was a strong correlation between the commanded and actual torques. A similar observation was made for the original actuator, with an experimentally tested peak torque of 17.4 Nm.

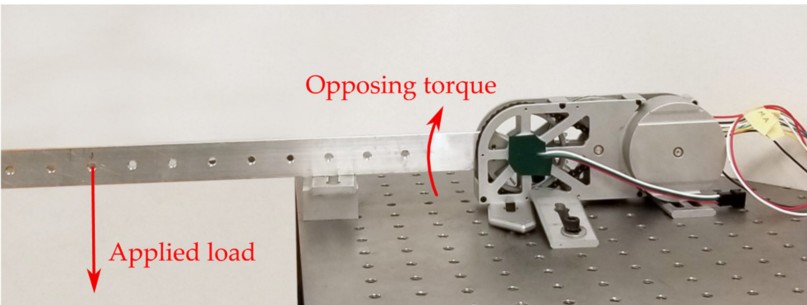

**Figure 3.** Experimental setup for the static torque test, showing the applied load on actuator arm and opposing actuator torque.

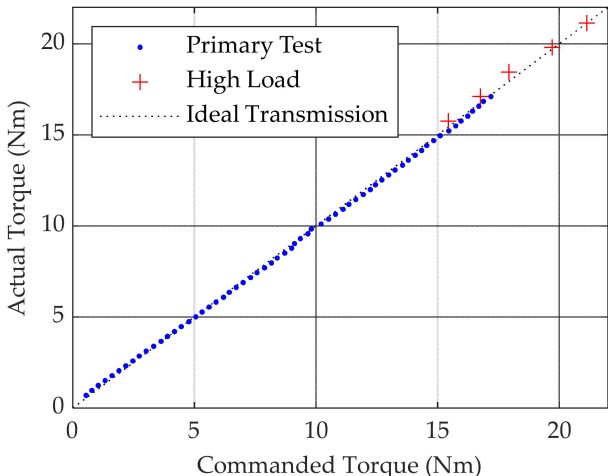

**Figure 4.** Static torque test, showing the actual vs. commanded torque from the actuator.

To complement the static torque test, an experiment was performed to identify the open-loop system. A sinusoidal commanded input torque was swept from 0.1 to 10 Hz, which encompasses the frequencies typically observed in gait [54], and the resultant output velocities were recorded via Hall effect sensors embedded in the motor and scaled by the actuator transmission ratio. The plant magnitude at each frequency was then computed based on the fitted sine wave amplitude and commanded torque amplitude. The frequency response data (FRD) are depicted in Figure 5a. The figure shows the second-order black-box model fitted to the amplitude data with the identified open-loop transfer function $H(s)$ included, relating input commanded torque to output velocity.

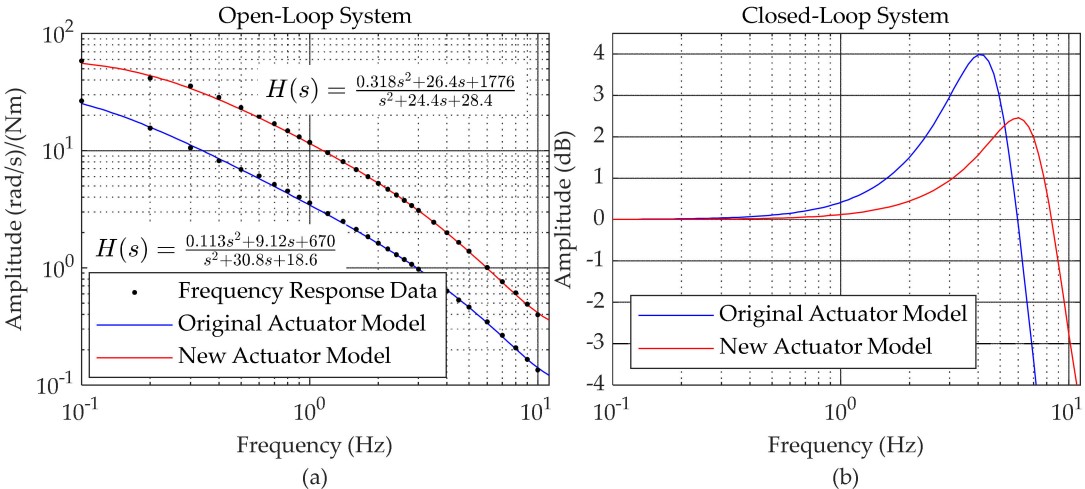

**Figure 5.** (**a**) Frequency response data and the identified actuator open-loop model from input torque to output velocity $H(s)$; (**b**) A proportional-derivative (PD) controller was used to create a closed-loop model and the closed-loop bandwidth was identified.

To find the closed-loop bandwidth of the actuator, a proportional-derivative (PD) state feedback controller was applied to the identified system for position tracking, yielding the closed-loop Bode diagram in Figure 5b. A simulated PD controller with a proportional gain of 35 Nm/rad and derivative gain of 2 Nm s/rad was used to calculate a bandwidth of 10.1 Hz. Greater closed-loop bandwidths are possible with larger PD gains, but the gains used in this calculation were selected to be identical to the controller used for gait tracking in Section 4.2. Although this does not represent an experimentally identified closed-loop bandwidth, the result does suggest that the actual bandwidth of the actuator is significantly greater than the frequencies found in gait. Following a similar procedure for the original actuator, a bandwidth of 6.8 Hz was calculated for identical PD gains.

### 3.2. Backdrivability

Backdrivability can be characterized in terms of the impedance of the system. Here, this incorporated inertia, viscous friction, Coulomb friction, and static friction. The system inertia was identified by assuming that net torque is proportional to angular acceleration with proportionality constant equal to the inertia. The servo-amplifier was configured to drive the actuator at a constant acceleration of 750 revolutions per minute per second (rpm/s) at the output. Torque data recorded during this time was used to calculate the system inertia as $1.45 \times 10^{-3}$ kg m/s$^2$. Following a similar procedure, the system inertia for the original actuator was larger at $2.84 \times 10^{-3}$ kg m/s$^2$.

For identifying viscous and Coulomb friction, the servo-amplifier was configured to drive the actuator at various constant velocities within their expected operating range (± 500 deg/s). The torque necessary to maintain the speed would primarily be due to viscous and Coulomb friction. The target speed was incrementally swept across this range in both directions, and measured torques were pooled from the data taken from operating in each direction. The recorded torques are portrayed in Figure 6, and a viscous and Coulomb friction model was fitted to the data using weighted least-squares linear regression with weights (*W*) defined by the square of the standard deviation. The parametric friction model and the weighted least-squares linear regression solution are

$$U = C\,sign(\dot{q}) + D\dot{q} \tag{1}$$

$$\Phi = \left(R^T W R\right)^{-1} R^T W U \tag{2}$$

where $\Phi = [C\ D]^T$, *C* is the Coulomb friction coefficient, *D* is the viscous friction coefficient, *U* is the input torques, $R = \left[sign(\dot{q}),\ \dot{q}\right]$ is the regressor matrix, and $\dot{q}$ is the recorded joint velocities.

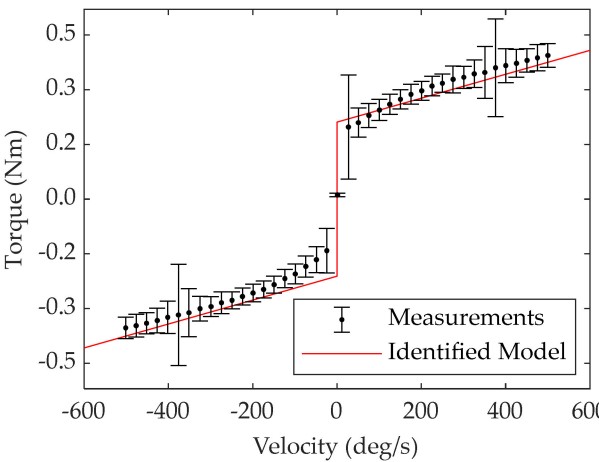

**Figure 6.** Measured friction levels with the actuator operating at various constant velocities. Error bars on the measured data represent one standard deviation from the mean. The identified model is superimposed, which incorporates Coulomb and viscous friction effects.

The identified viscous and Coulomb friction levels were 0.0188 Nm s/rad and 0.212 Nm, respectively. The viscous and Coulomb friction levels of the original actuator were identified as larger values at 0.0501 Nm s/rad and 0.685 Nm, respectively. Interestingly, the newly developed actuator exhibited a consistent increase in standard deviation when operational speed was 376 deg/s in either direction. The exact cause for this increase in variation is unclear, although it is suspected to be due to excitation of an internal resonance within the actuator. Furthermore, the actuator also showed a small increase in standard deviation at low velocities, possibly due to stick-slip friction from the hybrid belt–chain transmission.

For identifying static friction, the currents supplied to the motors were steadily increased from zero until a detectable velocity was recorded by the magnetic angle sensors at the output shaft.

This process was automated over two minutes for both directions. The recorded static friction is reported as an average between the two directions tested. The resultant static friction was greater than the Coulomb friction levels, at a value of 0.403 Nm. The static friction of the original actuator exhibited greater static friction levels of 0.730 Nm.

For a summary of the parameters characterizing backdrivability, see Table 4. The friction torque levels are all relatively low, representing a small proportion of the torque capabilities of the actuator. The newly developed actuator presented in this paper improved upon all metrics of backdrivability, as compared to the original version of the actuator.

**Table 4.** Inertia and friction parameters characterizing the backdrivability of the actuator.

| Actuator | Inertia | Viscous Friction | Coulomb Friction | Static Friction |
|----------|---------|------------------|------------------|-----------------|
| New | $1.45 \times 10^{-3}$ kg m/s$^2$ | 0.0188 Nm s/rad | 0.212 Nm | 0.403 Nm |
| Original | $2.84 \times 10^{-3}$ kg m/s$^2$ | 0.0501 Nm s/rad | 0.685 Nm | 0.730 Nm |

### 3.3. Speed and Operating Noise Tests

Actuator speed capability was tested to verify that the necessary joint speeds could be supported, with some margin for the eventual fluctuations that would be introduced when transitioning to an on-board portable power solution. Concurrently, sound output levels were measured, because quiet operation was a goal of the design, desired to make the device as approachable and non-intimidating as possible to the target pediatric population. The actuator without any load was tested in both directions up to 1176 deg/s, which is well above the maximum velocity dictated in Table 3. The actuator was placed 1 m from a microphone in the anechoic chamber at CSU and was run at a constant speed of 400 deg/s, which represents the maximum expected operating speed while in the exoskeleton, and operational noise was recorded for one minute. For comparison, one-minute recording of the empty anechoic chamber was also made to serve as a baseline for the noise levels. Although true decibel sound pressure levels were not obtained in this experiment, the setup did allow for relative decibel levels to be recorded and changes in the digital decibel levels were directly correlated to acoustic sound pressure levels. The amplitude spectrum of the noise recordings was calculated from the fast Fourier transform of the sound recordings and were A-weighted, which are reported in Figure 7. The operational noise level by the actuator was calculated as 43.3 dBA above the anechoic chamber baseline. For comparison, the original actuator was measured at 31.9 dBA above baseline.

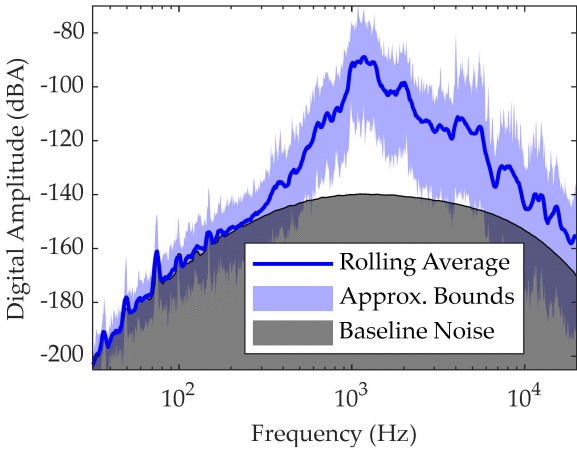

**Figure 7.** Amplitude spectrum of the audio signals recorded in the anechoic chamber. The rolling average shows the sound signal recorded in the frequency domain during actuator operation with approximate bounds to the recorded data included.

## 4. Evaluation on a Prototype Orthosis

As an initial performance evaluation, the actuators were integrated in a prototype orthosis to move a double pendulum system modified to exhibit mass, length, and inertial properties close to those of an average eight-year-old child, as performed in [35], and shown in Figure 8. A sine sweep experiment was conducted to gain an understanding of the system followed by a gait tracking experiment. The results of these experiments are detailed in the subsequent sections.

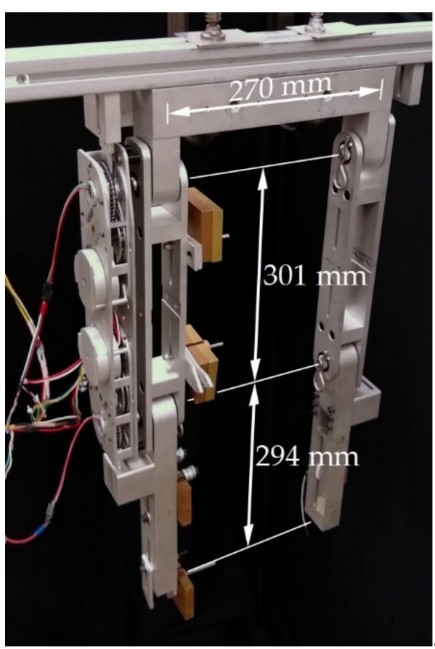

**Figure 8.** Photo of the actuators in the preliminary orthosis, attached to the pediatric test dummy leg.

### 4.1. System Response

A sine sweep experiment was conducted to identify an approximate linear model of the double pendulum system centered around the downward configuration depicted in Figure 8. An open-loop approach was taken, where inputs were the supplied torques and the outputs were the joint angles. The frequency of the supplied sinusoidal torque was swept from 0.25 to 4 Hz over 50 increments, which encompassed the bulk dynamics of the system. To ensure a high signal-to-noise ratio, the sinusoidal excitation signal amplitude was maximized at each increment. However, the amplitude was limited so that the resulting motion of the system would not hit the actuator's 30 deg hard limit in joint extension for both the hip and knee, and the peak current supplied to the motors did not exceed 4 A to ensure that the motor RMS current remained below the rated continuous current of 2.98 A. A sine wave was fitted to the resulting motion of each joint when exciting the hip and knee joints in two separate experiments, which were combined to create the FRD shown in Figure 9. A black-box linear state-space model was subsequently fitted to the FRD with four internal states, which is superimposed on the figure. The two natural frequencies identified by the model are 0.77 Hz and 1.59 Hz with damping ratios of 0.0433 and 0.232, respectively.

The frequency response diagram shows that the bulk dynamics of this system occur below 3 Hz, which is much lower than the theoretical closed-loop bandwidth of 10.1 Hz discussed in Section 3.1. It should be noted that at anti-resonant frequencies and other frequencies of low amplitude, the actuators may require higher torque to achieve satisfactory performance, which may not be possible due to current saturation from the motor drivers.

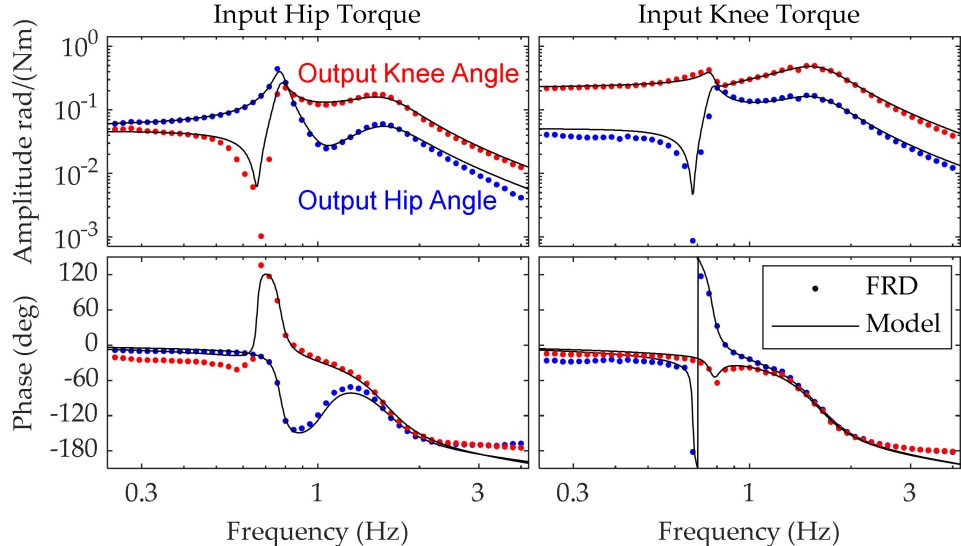

**Figure 9.** Frequency response data (FRD) and identified model from input torque to output position of the preliminary exoskeleton system on a prototype orthosis.

### 4.2. Gait Tracking with State-Feedback

To demonstrate the ability for the actuator to replicate a desired gait pattern, a state-feedback controller was created for gait tracking in the model system. Using the Winter gait data [51], a nominal gait pattern with a stride period of 1.1 s was created in the form of an 8-term Fourier series to generate a periodic function for gait reference at both the hip and knee. State-feedback took the form of PD control, where the gains were manually tuned until satisfactory gait tracking was attained. The PD gains and RMS values associated with the achieved gait tracking are listed in Table 5, based on the gait tracking results shown in Figure 10.

**Table 5.** Controller gains and summary of experimental results.

| Characteristic | Hip | Knee |
|---|---|---|
| Proportional Gain | 0.611 Nm/deg | 0.296 Nm/deg |
| Derivative Gain | 0.035 Nm s/deg | 0.019 Nm s/deg |
| RMS Angle Tracking Error | 3.79 deg | 6.33 deg |
| RMS Torque (% of CT) | 3.20 Nm (54.2%) | 2.70 Nm (45.7%) |
| Peak Torque (% of PT) | 7.39 Nm (34.9%) | 5.65 Nm (26.7%) |

CT = Rated continuous torque (5.9 Nm), PT = Tested peak torque (21.13 Nm).

The gains of the controller were kept low so as to not strain the hardware at this stage of evaluation. Furthermore, the experimenters aimed to keep the supplied RMS current at approximately 50% of the rated continuous current of the actuator. Subject to these constraints, the controller successfully achieved system motion resembling gait patterns for both the hip and knee joints. These preliminary results are promising, because they suggest that the actuators are capable of moving the system to follow gait patterns and may perform reasonably well in assisting a human user. The RMS and peak torques produced by the controller were a fraction of the continuous and tested peak torques, respectively, that the actuator can provide as shown in Table 5. This suggests that higher gains could be used to achieve better tracking, or other kinds of controllers could be applied to generate larger control inputs.

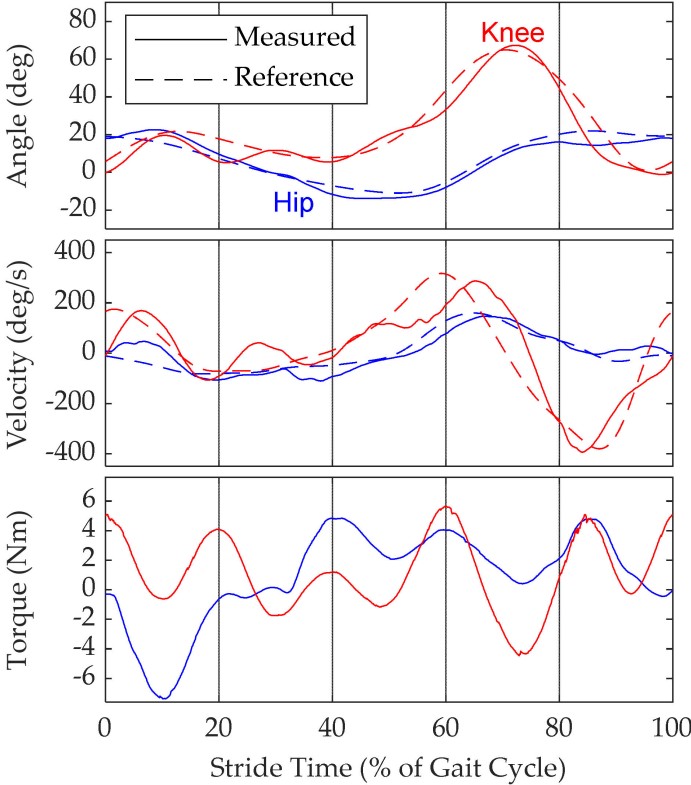

**Figure 10.** Experimental results for joint angle, velocity, and torque in gait tracking with the prototype orthosis and pediatric test dummy using state-feedback control.

## 5. Conclusions

This paper presented a newly developed joint actuator for an assistive pediatric lower-limb exoskeleton. The actuator is smaller and lighter than the original actuator previously presented. The specifications and design of the actuator were presented, and the actuator underwent a variety of tests to evaluate its capabilities which were contrasted with the original design. The torque and bandwidth were tested, the backdrivability was quantified in terms of inertia and friction, and speed and noise levels were assessed. The torque tests suggest that the actuators are able to accurately supply a commanded torque in a low-speed scenario. The bandwidth test indicates that the system is more responsive for the same amount of torque and a larger bandwidth than the original version. The inertia, viscous friction, Coulomb friction, and static friction levels are all lower than the original version of the actuator, indicating a drastic improvement in the backdrivability of the actuator. The speed tests show that the system can operate sufficiently fast enough to satisfy the design specifications. However, this comes at the cost of slightly higher noise levels than the original actuator, likely due to the exposed rotor in the frameless motor design. These results collectively demonstrate that the design specifications have been satisfied and an overall improvement in actuator performance over the original design. Additionally, the newly developed actuators were experimentally tested for their ability to perform gait tracking in a prototype orthosis with a pediatric test dummy. The satisfactory results suggest the viability of these actuators for use in the eventual pediatric exoskeleton being developed.

Several observations from the experiments should be made. The newly developed actuator draws a larger amount of current for the same amount of output torque produced than the original actuator. Given the terminal resistance of the actuators, the power loss from heat in the new actuator is larger than the original design by a factor of approximately two, suggesting that the new actuator may experience higher operating temperatures than the original actuator if continually strained. This has ramifications in the design of the pediatric exoskeleton where the actuator may rest close to the body. Future design iterations should investigate thermal performance as well as the inclusion of heat

sinks or active cooling if necessary. Additionally, in the static torque test, a minor but noticeable temperature dependency was observed between current and supplied torque, which can be further defined in future testing. The researchers also discovered some level of actuator backlash originating from the chain stage in the transmission. Although the minor backlash did not meaningfully affect most of the conducted experiments presented in this paper, it may have interfered with our attempts to perform a closed-loop bandwidth identification experiment. At high frequencies and subject to current constraints, the reference signal amplitude became comparable to the amount of actuator backlash observed, obscuring the bandwidth response. For the frequencies and joint amplitudes at which the actuator is expected to operate, the backlash effects should not have meaningful impact on performance.

All results considered, the newly developed actuator meets the design specifications and largely outperforms the original actuator. The results presented in this paper are promising, because they demonstrate that the actuators are suitable for use in a pediatric exoskeleton. Our group is continuing the development of a pediatric lower-limb exoskeleton utilizing these actuators in a validation study to evaluate the assistive capabilities of the CSU Pediatric Exoskeleton on patients in the near future.

**Author Contributions:** Conducted the experiments, A.G. and C.A.L.; composed the manuscript, A.G.; supervision, J.T.S.; review and editing, J.T.S and R.J.F. All authors collaborated in the design of the presented actuator. All authors have read and agreed to the published version of the manuscript.

**Funding:** This research received no external funding.

**Conflicts of Interest:** The authors declare no conflict of interest.

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
