# Peer review of "Design and Evaluation of a Pediatric Lower-Limb Exoskeleton Joint Actuator"

_actuators, doi:10.3390/act9040138_

Round 1
Reviewer 1 Report
The paper presents a joint actuator designed to drive the hip and knee joints of a pediatric lower-limb exoskeleton. The developed actuator incorporate a hybrid belt-chain transmission driven by a frameless brushless DC motor.
Variants of such rehabilitation systems are known in literature. Hence the novelty of this work is quite limited. The constructive solution presented in the paper is a fairly simple one, its innovative character being questionable.
The paper layout is generally correct and clear.
The authors may consider the following comments for revising the paper:
- In Table 1 the information concerning Joint velocity [deg/s]: aren’t the specified values too high? In systems designed for persons with disabilities shouldn’t the motions occur at slower speeds?
- The same in Table 3, isn’t the speed 370 deg/s too high?
- At page 8, row 241: “…a constant acceleration of 750 rpm/s…”. What does rpm/s mean?
- At page 8, row 242: “1.45×10−3 kg-m/s2”. What is the significance of the “-“ sign between kg and m? The same in the following rows in the text and in Table 4.
- In Fig. 8 information should be included about the distances between the rotation axes of the joints.
Reviewer 2 Report
The present manuscript presents the novel design and evaluation of a pediatric lower-limb exoskeleton joint actuator. The proposed actuator incorporate a hybrid belt-chain transmission driven by a frameless brushless DC motor. The dynamic performance of the actuator is evaluated. Moreover, the application of the actuator in a prototype orthosis is also presented to show the performance of the actuator. The actuator meets the design specifications and it is suitable for use in the pediatric exoskeleton.
The paper is well written. The following comments are suggested in order to improve the quality of the paper:
1) The equation of the actuator model is not presented; Nevertheless, the numerical simulation based on the model was used to evaluate the performance of the actuator. It could be interesting to present the equations of the actuator model and the definition of the parameters.
2) additional details about the frequency response function of Fig. 5 should be added: what are the input and output consider for H(s)? how was implement the PD controller based on the dynamic model of the actuator of Fig. 1? The selection of the gains modifies the frequency response. Is it possible to analyze H(s) for several PD gains? How was the experimental procedure to obtain the frequency response data (sensors, inputs.. etc)?
3) Were the Coulomb and viscous friction identified by using the least square method? Additional details should be added.
4) The results of Fig. 9 were described. A conclusion to discuss how this frequency response affects the performance of the actuator should be added.
5) Figures: the size of the labels could be increased to improve the readability.
